# Three-Stream and Double Attention-Based DenseNet-BiLSTM for Fine Land Cover Classification of Complex Mining Landscapes

**Diya Zhang** [1], **Jiake Leng** [1], **Xianju Li** [1,2,*], **Wenxi He** [3] **and Weitao Chen** [1,2]

1   Faculty of Computer Science, China University of Geosciences, Wuhan 430074, China
2   Hubei Key Laboratory of Intelligent Geo-Information Processing, China University of Geosciences, Wuhan 430074, China
3   Wuhan Centre of China Geological Survey, Wuhan 430205, China
*   Correspondence: ddwhlxj@cug.edu.cn

**Abstract:** The fine classification of land cover around complex mining areas is important for environmental protection and sustainable development. Although some advances have been made in the utilization of high-resolution remote sensing imagery and classification algorithms, the following issues still remain: (1) how the multimodal spectral–spatial and topographic features can be learned for complex mining areas; (2) how the key features can be extracted; and (3) how the contextual information can be captured among different features. In this study, we proposed a novel model comprising the following three main strategies: (1) design comprising a three-stream multimodal feature learning and post-fusion method; (2) integration of deep separable asymmetric convolution blocks and parallel channel and spatial attention mechanisms into the DenseNet architecture; and (3) use of a bidirectional long short-term memory (BiLSTM) network to further learn cross-channel context features. The experiments were carried out in Wuhan City, China using ZiYuan-3 imagery. The proposed model was found to exhibit a better performance than other models, with an overall accuracy of 98.65% ± 0.05% and an improvement of 4.03% over the basic model. In addition, the proposed model yielded an obviously better visual prediction map for the entire study area. Overall, the proposed model is beneficial for multimodal feature learning and complex landscape applications.

**Keywords:** remote sensing; CNN; LSTM; attention mechanism; multistream; multimodal; land cover classification

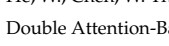

## 1. Introduction

The illegal over-exploitation of open pit mines can easily lead to surface subsidence and desertification, which aggravates soil erosion and vegetation reduction. Soil erosion and reduced vegetation can lead to changes in the surface land cover types. According to recent studies, changes in surface land cover types have a significant impact on the surface temperature of the region. Furthermore, changes in the surface temperature increase with changes in the surface land cover [1]. The increase in surface temperature creates a series of environmental problems that affect the goal of sustainable development.

Real-time monitoring of open pit mining activities is imperative. An effective approach for monitoring the activities of open pit mines involves the detection of changes in land cover types in and around the mine. The fine classification of the different types of land cover in mining areas and their vicinity is of great significance for environmental protection, sustainable development, and scientific mining. The detection of land cover is an important method for reflecting the ecological environment [2], and it can effectively prevent the threat of dust, environmental pollution, and vegetation damage caused by mineral resource development. However, owing to the complex topography near the open pit mine and damage to the land by mining activities, performing manual observations of the complex open pit mining areas and detecting their changes in real-time are impractical [3].

Multispectral images provide rich spectral information and have broad application prospects in the field of remote sensing [4,5]. In recent years, with the development of multispectral imaging technology, the coverage of remote sensing images has increased [6], and the spatial resolution of remote sensing images has continuously improved [7,8]. However, such advancements have increased the information complexity of multispectral remote sensing images [9–11] and has led to a reduction in the spectral differences between different types of objects. The spectral difference within the class increases [12,13], and the spectral–spatial information cannot be fully utilized [14,15]. Because open pit mining areas have complex landscapes and significant three-dimensional topographic features [16,17], the gathering of multimodal spectral–spatial and topographic information to improve the land cover classification (LCC) accuracy of complex open pit mining areas is an issue.

Friedl and Brodley [18] used three decision tree algorithms (i.e., univariate, multivariate, and mixed decision trees) to conduct LCC studies. In a prior study [19], the researchers performed pixel- and object-based experiments on MS images acquired by the HJ-1B and ALOS satellites using the support vector machine (SVM) and random forest (RF) algorithms, respectively. Machine learning algorithms (MLAs) are effective for LCC and fine LCC (FLCC) applications. However, MLAs generally have shortcomings when classifying remote sensing images in complex landscapes. Due to the utilization of feature engineering with a low representation ability, MLAs cannot fully extract image information [15]. As a result, the corresponding classification accuracies are unsatisfactory.

In the past few years, deep learning (DL) algorithms have been widely used for FLCC applications due to their powerful feature learning abilities [20,21], and they have achieved good results in data mining, natural language processing, and other fields [22]. The restricted Boltzmann machine in the deep belief network (DBN) can be trained while unsupervised to learn the features that the original data contain to the greatest extent; while supervised, it can be fine-tuned using the back-propagation algorithm to further learn data features, which requires little labeled data for the FLCCs of complex mining areas [23]. Li et al. [24] proposed a multi-level output-based DBN model for FLCCs in complex mining areas. Helber et al. [25] used a deep convolutional neural network (DCNN) to perform an LCC of Sentinel-2 satellite images. Xu et al. [26] used a multiscale DCNN to achieve good classification results. However, the DCNN method has a deep network structure and numerous model parameters. Accordingly, gradient disappearance and network degradation can easily be induced. The ResNet architecture increases the network depth through fast identity mapping [27], thereby improving the feature extraction ability and avoiding the above-mentioned issues of the DCNN method. However, ResNet connects the input and residual feature maps by summing them in the skip connection stage, which may block the information flow. ResNet also generates numerous feature maps in each layer and trains with too many parameters, leading to excessive training time. These deficiencies can be solved by the DenseNet architecture [28], as each layer of DenseNet is connected to all subsequent layers in the forward feedback mode, and its input includes the output of the previous layer and the input of all layers before the current output layer. The shallow features that are directly extracted from the original data can be directly used by the deep layers; therefore, DenseNet enhances feature reuse, reduces the number of parameters, and achieves stronger performance than the ordinary CNN method. Therefore, many researchers have introduced it into the field of remote sensing. Tao et al. [29] added a classifier at a certain interval between the input and output layers in the DenseNet architecture to improve the feature extraction ability of the network. Li et al. [30] inputted the features of the last dense block of DenseNet121 into the global branch of the key filter bank to obtain global features; the authors also adopted the right branch of the key flow to extract features from key regions, improving the classification accuracy of hyperspectral imagery (HSI).

Multimodal data-based multistream CNNs are widely used in the field of remote sensing. For example, Wu et al. [31] designed a dual-stream CNN to extract the features of HSI and light detection and ranging (LiDAR) data. Hang et al. [32] designed a dual-stream

CNN structure with mutually coupled convolutional layers to extract and fuse HSI and LiDAR features.

Multispectral remote sensing images have spectral features that are related to different spectral bands as well as spatial features that are related to different neighborhoods. Making full use of the spectral spatial features simultaneously is key to improving the classification accuracy. Xu et al. [26] used a three-dimensional CNN to mine spectral–spatial features. To further improve the classification accuracy of the HSI, Zhao and Du [33] designed a multiscale convolutional network to build an image pyramid for each spectral band by fusing the extracted cross-scale spatial features with spectral features. Similarly, Gao et al. [34] designed a deep cross-attention module to extract complementary information from spectral–spatial features. Researchers also introduced making use of generative adversarial networks (GANs) to mine spectral features. Li et al. [35] proposed a deep translation-based change detection network (DTCDN) to automatically extract the variation area between optical and synthetic aperture radar remote sensing images. To establish a more fluent and stable training, Zhang et al. [36] optimized the training process and loss function for GANs and proposed a hyperspectral image classification optimization training method based on GAN. The proposed method can obtain good training results with only a small amount of labeled training data.

Most existing spectral–spatial fusions are based on adjacent pixel feature extraction methods, which do not maximize the contextual information of the spatial–spectral features. A long short-term memory (LSTM) network can establish interdependence between the input sequences and is widely used in natural language processing, video processing, machine translation, and other fields [37]. In the past few years, this particular network has demonstrated excellent performance in remote sensing image processing. Xu et al. [38] designed a band grouping strategy that enabled LSTM networks to better learn the contextual information between adjacent channels of the HSI. To obtain combined spectral–spatial results, researchers have proposed a spectral–spatial LSTM model. Zhou et al. [39] designed a spectral–spatial LSTM to extract the spectral–spatial features of the HSI, which captured the dependencies of adjacent spectral–spatial neighborhoods. Liu et al. [12] proposed a spectral–spatial bidirectional LSTM (BiLSTM) network to simultaneously extract spectral–spatial features using recursive and convolution operators. Similarly, Yin et al. [40] used a 3D CNN to extract spatial features and a band grouping strategy-based BiLSTM to extract spectral features.

Although DCNNs have a strong feature extraction ability, their ability to extract key features in a complex landscape is not strong enough. The attention mechanism can focus on the different aspects of each output of the inputs, thereby improving its ability to extract the most relevant information. Chen et al. [41] proposed a framework based on global context spatial attention and the DenseNet architecture to extract multiscale global scene features. Some researchers are beginning to determine the effects of different attention mechanisms on key feature extraction. Chen et al. [42] proposed generative adversarial networks based on the joint mechanism of channel and spatial attention for land cover classification, which assigns higher weights to more important features and reduces the interference of invalid features. Tong et al. [43] proposed a channel attention-based DenseNet method for scene classification. The channel attention mechanism adaptively strengthens the weights of the important feature channels and suppresses the secondary feature channels. To represent the importance weights of different local regions of each image, Sumbul and Demİr [44] proposed a novel multi-attention strategy-based BiLSTM network to define a global descriptor for each image.

In summary, three issues exist in the FLCCs of complex mining areas: (1) how the multimodal spectral–spatial and topographic features can be learned, (2) how the key features can be extracted, and (3) how the contextual information can be captured among different features. To improve the classification accuracy of FLCCs in complex mining areas, this study proposed a three-stream, double attention network (3S-A$^2$ DenseNet-BiLSTM). This network comprises three strategies: (1) A three-stream multimodal feature

learning and post-fusion method. Briefly, the multispectral imagery and topographic data were first fed into subsequent DL networks. Thereafter, the low-level spectral–spatial features were fused with the extracted multimodal deep features; (2) An integration of deep separable asymmetric convolution blocks (ACB-DS), parallel channel, and spatial attention mechanisms into the DenseNet architecture for key feature extraction; and (3) the use of the BiLSTM network to further learn the cross-channel context features from the outputs of the $A^2$ DenseNet. The proposed model was tested in Wuhan City, China, using Ziyuan-3 (ZY-3) stereo satellite imagery.

The remainder of this paper is organized as follows: Section 2 introduces the study area and data sources; Section 3 describes the proposed network structure, feature fusion strategy, parameter optimization, and evaluation indicators; Section 4 presents the analysis of the experimental results, including the experimental equipment, parameter optimization results, and accuracy evaluation; Section 5 discusses the effectiveness of the model and compares it with models used in previous research; and Section 6 presents the conclusions.

## 2. Study Area and Remote Sensing Datasets

In this study, the same research area was adopted as previous studies [7,23,24], which is located in the Wulongquan mining area, covering an area of 109.4 km$^2$ in the Jiangxia District, Wuhan City, Hubei Province, China. The area is a large ore production base and the mining resources in this area are distributed in contiguous and concentrated areas. Different complex open pit mining areas and agricultural development landscapes exist at different phenological stages.

At present, commonly used satellite images include Sentinel-3, Landsat, Sentinel-2, and HJ-1A [45]; however, ZY-3 satellite images are widely used in FLCCs due to their higher spatial resolution. According to prior studies, the higher the spatial resolution is in multispectral satellites, the more accurate the classification of land cover types in the region [46]. Figure 1 shows a satellite image captured by the ZY-3 satellite on 20 June 2012. As the front- and back-facing cameras had a resolution of 3.6 m, the digital elevation model (DEM) data had a resolution of 10 m. The nadir-looking panchromatic camera had a resolution of 2.1 m, and the multispectral camera had a resolution of 5.8 m. We used multispectral imagery from an optical satellite in this study as the type of satellite images (that is, the ZY-3 satellite). The four bands of blue visible light, green visible light, red visible light, and near-infrared light, which were fused to form the multispectral imagery by Gram–Schmidt spectral sharpening; a fused multispectral imagery with a resolution of 2.1 m was obtained. Subsequently, the DEM was resampled to a 2.1 m resolution to match the multispectral imagery.

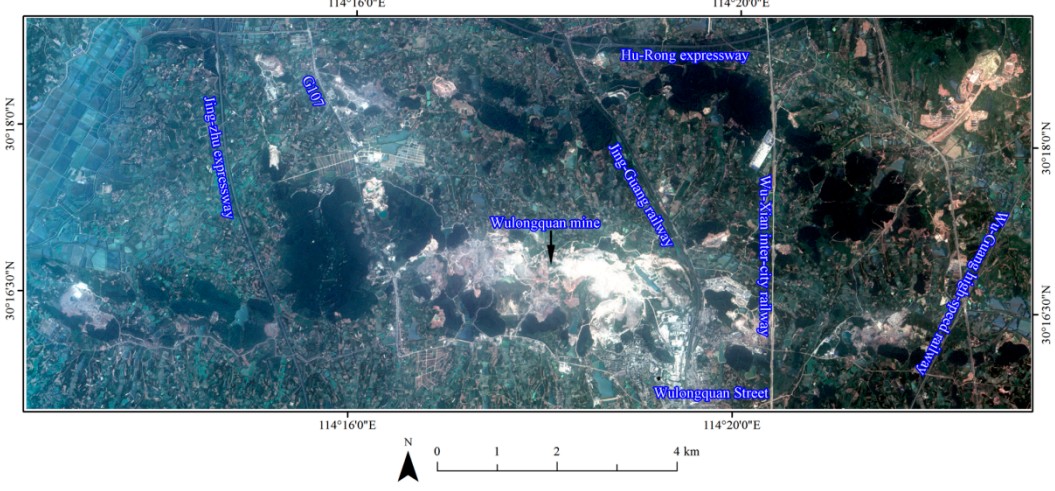

**Figure 1.** ZY-3 satellite remote sensing image of the study area (revised from [23]).

In this study, a two-level land cover classification system was designed [47] using twenty-eight field survey samples, including seven first-level and twenty second-level land covers, as shown in Table 1. The second-level land cover type is a detailed classification of the first-level land cover type. The FLCC was based on the second-level land cover type. The design of the two-level land cover classification system is conducive to the classification of land cover on a fine scale.

**Table 1.** The two-level land cover classification system [47].

| First-Level Land Cover Type | Second-Level Land Cover Type | Description |
| --- | --- | --- |
| Mining land | Open pit | An ore deposit formed by stripping the soil and rock covering the upper part of the ore body |
| | Ore processing site | A factory that processes ore |
| | Dumping ground | A place where mining wastes are discharged in a centralized manner |
| Farmland | Paddy | Cultivated land for planting rice |
| | Greenhouse | Facilities that can transmit light and keep warm, which are used to cultivate plants |
| | Green dry land | Cultivated land not planted with rice |
| | Fallow land | Uncultivated land |
| Woodland | Woodland | Tall macrophanerophytes |
| | Shrubbery | Low vegetation |
| | Coerced forest | Vegetation with restricted growth |
| | Nursery | Economic trees artificially cultivated in nurseries and orchards |
| Waters | Pond and stream | Natural waters |
| | Mine pit pond | A place where groundwater is discharged during mining |
| Road | Dark road | Asphalt driveway |
| | Light gray road | Dirt road |
| | Bright road | Cement road |
| Residential land | Blue roof | Urban land |
| | White roof | Land for rural residents |
| | Red roof | Other construction land |
| Unused land | Bare surface land | |

Based on our previous studies [7,23,24,47,48], the training, validation, and test sets were constructed using data polygons. Each class of the training set contained 2000 samples, and there were 40,000 in total. Each class of the validation set had 500 samples, 10,000 in total; the test set had the same number of samples. The sample proportion of the training, validation, and test sets was at a ratio of 4:1:1, respectively. The training set was used to fit and train multiple classification models. To identify the model with the best performance, each trained model was used to predict the validation set, and we selected the model with the best classification effect. The test set was used to evaluate the model's accuracy.

## 3. Methods

### 3.1. Description of the Proposed 3S-A$^2$ DenseNet-BiLSTM

As shown in Figure 2, the neighborhoods of the $N \times N$ pixels were first extracted from the multispectral image and topographic data, respectively, and the neighborhood size was subjectively determined according to the actual situation. Thereafter, these pixel neighborhoods were inputted into the A$^2$ DenseNet-BiLSTM network using an asymmetric separable convolution block and double attention mechanism to extract the cross-channel and context-dependent information. The two branches adopted a parameter sharing method to learn network weights and extract similar features at the same location of different input data, that is, the deep spectral–spatial and topographic features. Subsequently, the low-level spectral–spatial features extracted from the multispectral images were fused with the depth features and inputted into Softmax for classification. By comparing the

prediction results with the real labels, the batch size loss was calculated according to the loss function. Finally, a back-propagation algorithm was used to optimize the model.

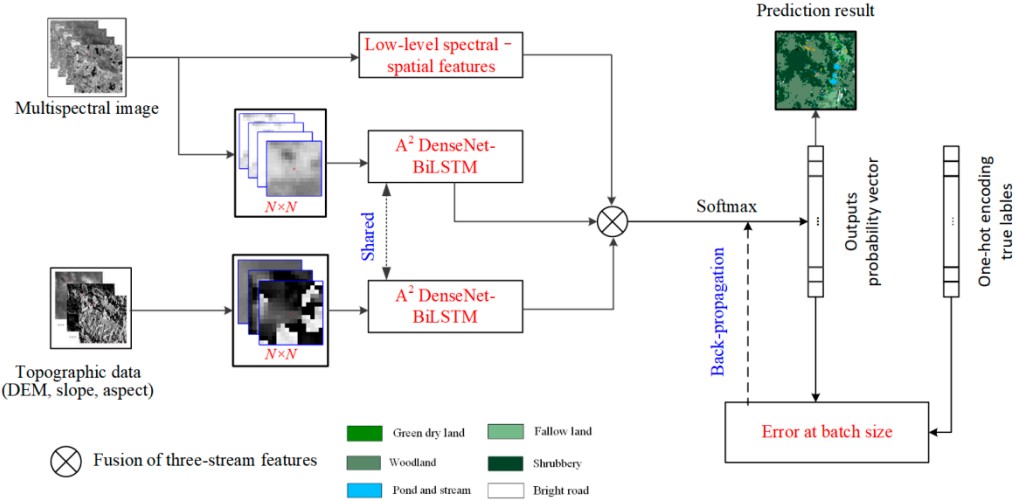

**Figure 2.** The algorithm flow chart of the 3S-A$^2$ DenseNet-BiLSTM.

### 3.2. A$^2$ DenseNet for Key Feature Extraction

The A$^2$ DenseNet (Figure 3a) network includes the ACB-DS block (Figure 3c), dense block (Figure 3d), convolution block, and transition layer. The dense connection mechanism of DenseNet connects each layer in the dense block, and the input of each layer is the output of all previous layers. If the dense block has *L* layers, the number of connections can reach $\frac{L(L+1)}{2}$. The nonlinear transformation function is $H_L(\cdot)$, and the output of the *L* layer is $X_L$. For a standard CNN, the output of the *L* layer is given by:

$$X_L = H_L(X_{L-1}) \tag{1}$$

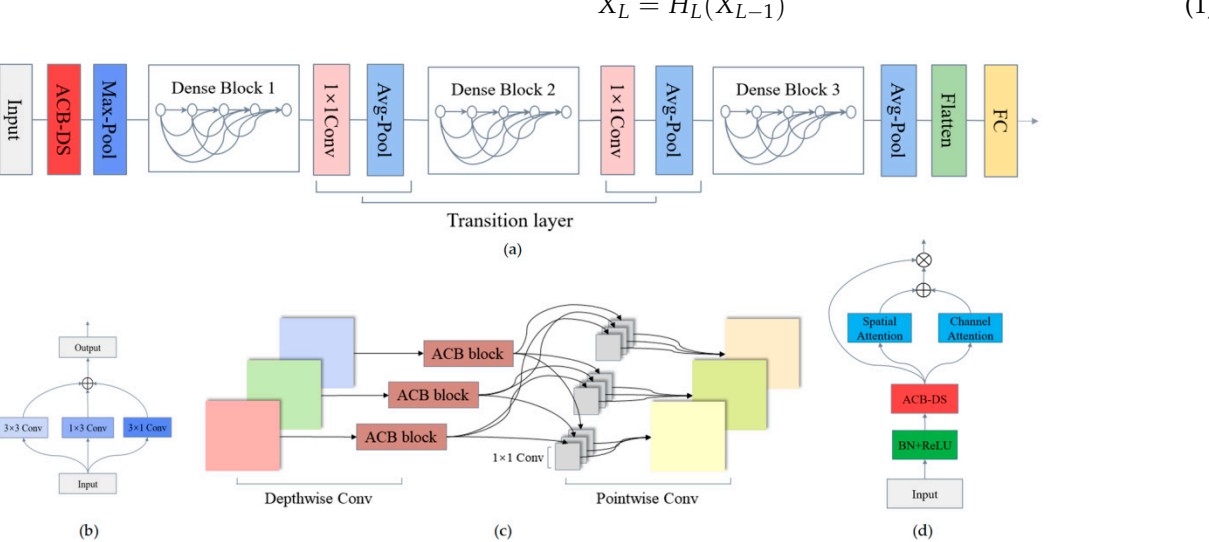

**Figure 3.** Structure of the A$^2$ DenseNet network (**a**), asymmetric convolution block (ACB) (**b**), ACB-DS block (**c**), and dense block (**d**).

For the dense block, the output of its *L* layer is given by:

$$X_L = H_L([X_0, X_1, \cdots, X_{L-1}]) \tag{2}$$

The deeper the layers of input data and gradient information transfer, the easier the occurrence of the network in the gradient disappearance phenomenon. Such a dense connection mechanism allows for a more effective transfer of features and gradients and

alleviates the problem of gradient disappearance caused by the network being too deep. This mechanism has a regularizing effect that reduces the computational parameters in the network and alleviates the overfitting phenomenon [43].

As shown in Figure 3a, each dense block in DenseNet is followed by a transition layer that consists of a $1 \times 1$ convolution and average pooling operation. The $1 \times 1$ convolution can reduce the number of output feature maps and fix the number of channels in output feature maps to reduce the image dimensions, reduce the number of parameters in the network, and reduce the model complexity.

The $H_L(\cdot)$ structure of the classic DenseNet comprises BN, ReLU, and convolution, with a kernel of $3 \times 3$. In this study, ACB-DS (Figure 3c) was used to replace the original convolution, which included ACB and deep separable (DS) convolution. For ACB (Figure 3b), the output feature map is the sum of the input images after $3 \times 3$, $1 \times 3$, and $3 \times 1$ convolutions, which are equivalently fused into a square convolution kernel to replace a single convolution block. ACB strengthens the feature representation and feature extraction capabilities of the convolution layer, making the extracted features more discriminative and feature extraction more effective [49].

The channel attention (Figure 4a) and spatial attention (Figure 4b) mechanisms were introduced into the dense block (Figure 3c). These mechanisms were embedded in a parallel form, where the channel attention was implemented by a squeeze-and-excitation net (SENet). An SENet focuses on the inter-channel relationship and explicitly models the inter-channel interdependence to adaptively recalibrate the channel feature response [50]. This network includes two main operations: squeezing and excitation. For the squeeze operation, an adaptive average pooling operation was performed on the spatial dimension. During the compression of the spatial dimension, the average value of the corresponding dimension can be obtained, which can suppress unnecessary features. According to the structure in Figure 4a, the size of the original feature map is $H \times W \times C$, denoted as $u_c$, where $H$ and $W$ denote the spatial dimension of the feature map and C is the number of channels. The squeeze operation compresses the feature map $H \times W \times C$ into a one-dimensional feature vector of $1 \times 1 \times C$, denoted as $z_c$, which is obtained as follows:

$$z_c = F_{squeeze}(u_c) = \frac{\sum_{i=1}^{H} \sum_{j=1}^{W} u_c(i,j)}{H \times W} \tag{3}$$

For the excitation operation, a multilayer perceptron was first used to learn the channel weight, which had two layers: a fully connected (FC) layer with the activation function of ReLU (represented by $\delta$) and a FC layer with a sigmoid function (represented by $\sigma$). Additionally, $s$ denotes the output weight matrix, $g$ denotes the first FC layer, and $F_{excitation}$ denotes the second FC layer. The relations among them are expressed as follows:

$$s = F_{excitaion}(z_c) = \sigma(W_2 \delta(W_1 z_c)) \tag{4}$$

The obtained weight was multiplied by the original feature map to obtain the final weighted feature map; this was performed to learn the features that were useful for the current task.

The spatial attention mechanism (Figure 4b) can cause the network to focus more on the spatial location of important features. First, the maximum pooling and average pooling operations were performed on the input feature map (represented by $F$) of $H \times W \times C$. Thereafter, the two feature maps were cascaded according to the channel. Finally, the convolution operation $f$ with activation function $\sigma$ was conducted as follows:

$$M(F) = \sigma(f(AvgPool(F), MaxPool(F))) \tag{5}$$

The spatial attention mechanism can enhance the spatial structure features and spatial neighborhood-related information [51]. The channel and spatial attention branches were merged and fused. The fused feature map has more discriminative spectral–spatial features.

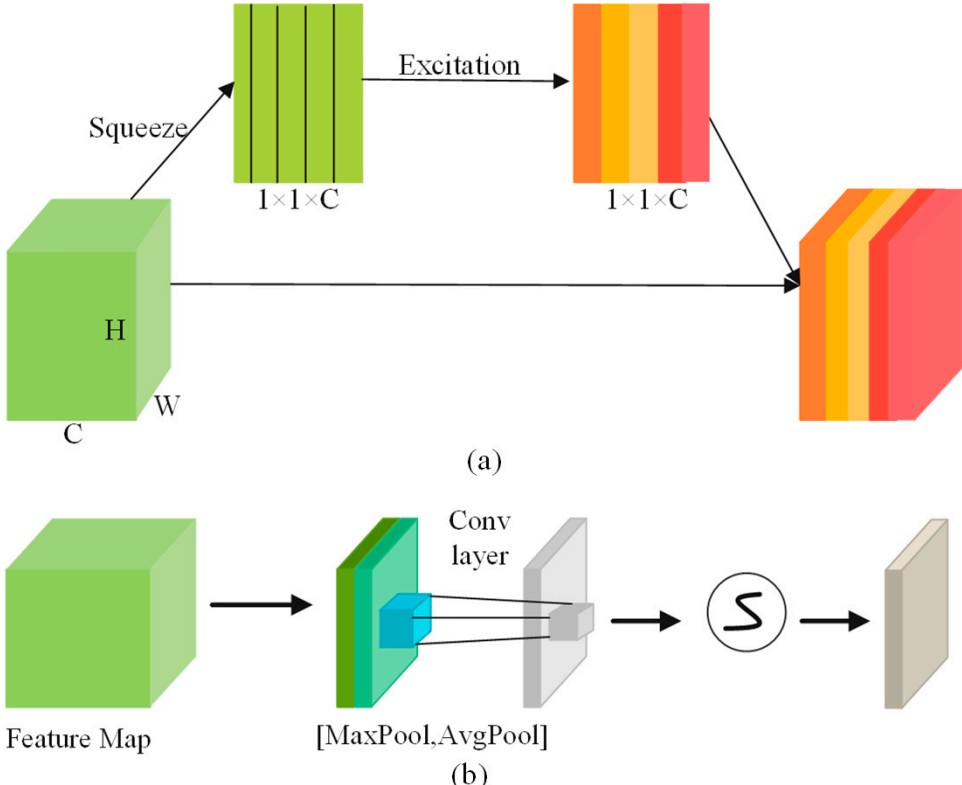

(a)

(b)

**Figure 4.** Structure of the channel attention module (**a**) and spatial attention module (**b**).

### 3.3. BiLSTM for Extraction of Cross-Channel Context Features

LSTM is a special recurrent neural network (RNN). Back-propagation and long-term dependencies easily lead to RNN gradient disappearance and explosion. Use of an LSTM introduces a gate mechanism and memory unit to improve RNN [52].

$C_t$ denotes the memory unit of the LSTM and memorizes all sequence information up to time $t$, $\widetilde{C}_t$ denotes the update value of the memory cell, $x_t$ denotes the input information at time $t$, $b$ denotes the offset value, $W$ denotes the weights between the LSTM nodes, and $h_t$ denotes the hidden layer output. The forget gate $f_t$ controls the forgetting of historical information, the input gate $i_t$ controls the stored information, and the output gate $o_t$ controls the output information. The mathematical illustrations of the LSTM information flow are given by:

$$f_t = \sigma\left(W_f \cdot [h_{t-1}, x_t] + b_f\right) \tag{6}$$

$$i_t = \sigma(W_i \cdot [h_{t-1}, x_t] + b_i) \tag{7}$$

$$\widetilde{C}_t = tanh(W_C \cdot [h_{t-1}, x_t] + b_C) \tag{8}$$

$$C_t = f_t * C_{t-1} + i_t * \widetilde{C}_t \tag{9}$$

$$o_t = \sigma(W_o \cdot [h_{t-1}, x_t] + b_o) \tag{10}$$

$$h_t = o_t * tanh(C_t) \tag{11}$$

In this study, the feature map $X \in R^{H \times W \times C}$ extracted by the $A^2$ DenseNet ($H \times W$ is the feature map size and C is the number of channels) was flattened and fully connected to obtain a one-dimensional feature vector. The feature sequence corresponding to each spectral band was inputted into the BiLSTM (Figure 5). Accordingly, the $[1, C]$ group sequence $\left\{X^{(1)}, X^{(2)}, \cdots, X^{(C)}\right\}$ was employed as the input of the BiLSTM module to extract the context information across channels. The use of BiLSTM can not only establish the forward correlation between the input sequences, but also learn the dependency between them in the backward sequence to fully extract the context information between different channels.

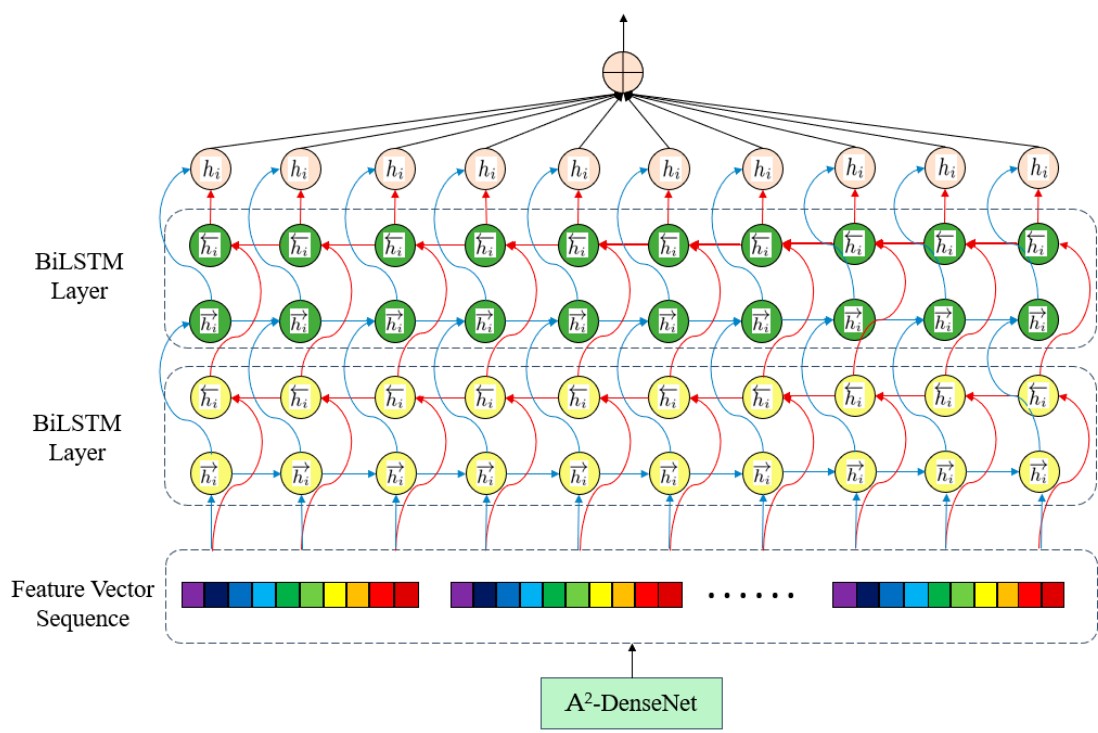

**Figure 5.** Structure of BiLSTM.

### 3.4. Three-Stream Multimodal Feature Learning and Post-Fusion Strategy

The multispectral data included four channels with a neighborhood size of $16 \times 16 \times 4$, and the topographic data included three channels with an input size of $16 \times 16 \times 3$. Based on the multispectral imagery and DEM data, six types of spectral–spatial and topographic features were extracted (Table 2), and the feature dimension was 106. The topographic data included the DEM, slope, and aspect. There were 103 low-level spectral–spatial features. Based on the $A^2$ DenseNet-BiLSTM results for the deep features from the multispectral imagery and topographic data branches, a post-fusion was conducted with low-level features.

**Table 2.** Low-level multimodal features.

| Feature Parameter Type | Feature Parameter Name | Number |
|---|---|---|
| Spectral features | Spectral bands | 4 |
| Principal component features | First and second principal components of spectral bands | 2 |
| Vegetation index | Normalized vegetation index eliminating, with the difference between the two channel reflectors | 1 |
| Filter features | Gaussian low-pass, mean, and standard deviation filtering in the spectral band with a core size of $3 \times 3$, $5 \times 5$, and $7 \times 7$ pixels | 36 |
| Texture features | Gray level co-occurrence matrix texture in the spectral band, including the contrast, correlation, angular second moment, homogeneity and entropy. Core size $3 \times 3$, $5 \times 5$, and $7 \times 7$ pixels | 60 |
| Topographic features | DEM, slope, aspect | 3 |

### 3.5. Model Construction and Parameter Optimization

This study sought to design three groups of comparative experiments and one group of the overall experiment.

Comparative Experiment 1: Feature fusion method based on multispectral images, topographic data, and two DenseNet branches.

Comparative Experiment 2: Feature fusion method based on multispectral images, topographic data, and two $A^2$ DenseNet branches.

Comparative Experiment 3: Feature fusion method based on multispectral images, topographic data, and two $A^2$ DenseNet-BiLSTM branches.

Overall Experiment: The proposed 3S-$A^2$ DenseNet-BiLSTM. The three-stream feature fusion methods are based on multispectral images, topographic data, and low-level spectral–spatial features. In addition, there are two $A^2$ DenseNet-BiLSTM branches.

For the four sets of comparative experiments, a total of 4 types of parameters were optimized in this study, as shown in Table 3. Some parameters were optimized by the trial-and-error method and included a batch size of 64, input size of $16 \times 16$, dense block number of three, BiLSTM layer of two, and $A^2$ DenseNet growth rate of 32.

**Table 3.** Parameter optimization scheme.

| Comparative Experiment 1 | | Other Experiments | |
| --- | --- | --- | --- |
| | DenseNet121 | ACB-DS | 1, 2, 3, 4 |
| DenseNet depth | DenseNet161 | Convergent epoch | 1–200 |
| | DenseNet169 | Fully connected layer | 1, 2, 3 |
| | DenseNet201 | | |
| Convergent epoch | 1–200 | | |
| Fully connected layer | 1, 2, 3 | | |

### 3.6. Accuracy Assessment Metrics

In this study, four accuracy evaluation metrics, namely the overall accuracy (OA), kappa, F1-score, and F1-measure, were selected to evaluate the performance of the proposed models. The F1-measure was used to assess the accuracy of each land cover type. The F1-score was the average value of all F1-measures.

## 4. Results

### 4.1. Results of Parameter Optimization

The four sets of models ran on a machine with the Centos 7 operating system and two NVIDIA 1080Ti GPUs with 11 GB of memory.

Five sets of experiments were repeated five times on the training and validation sets, each with 200 iterations. The results were averaged, and 12 sets of parameters (Table 3) were optimized for each set of experiments. The optimization results of the four experimental groups are shown in Figure 6. The Overall Experiment achieved the best effect when the numbers of the ACB-DS blocks and FC layers in the dense block were set to 1, and when DenseNet121 was used. The corresponding average validation accuracy reached 98.85% $\pm$ 0.08%. Comparative Experiment 1 achieved the best effect when DenseNet201 was used, and the numbers of the ACB-DS blocks and FC layers were 4 and 1, respectively. The average validation accuracy of Comparative Experiment 1 reached 93.70% $\pm$ 1.23%. Comparative Experiment 2 achieved the best effect when using the same parameter combination as that of the Overall Experiment. The average validation accuracy reached 97.71% $\pm$ 0.09%. Comparative Experiment 3 also achieved the best effect when using the same parameter combination as that of the Overall Experiment. The corresponding accuracy was 97.69% $\pm$ 0.27%.

As shown in Figure 7, as the epoch increased, the training and validation accuracies of the four sets of experiments displayed an upward trend, and the losses gradually decreased in a small range.

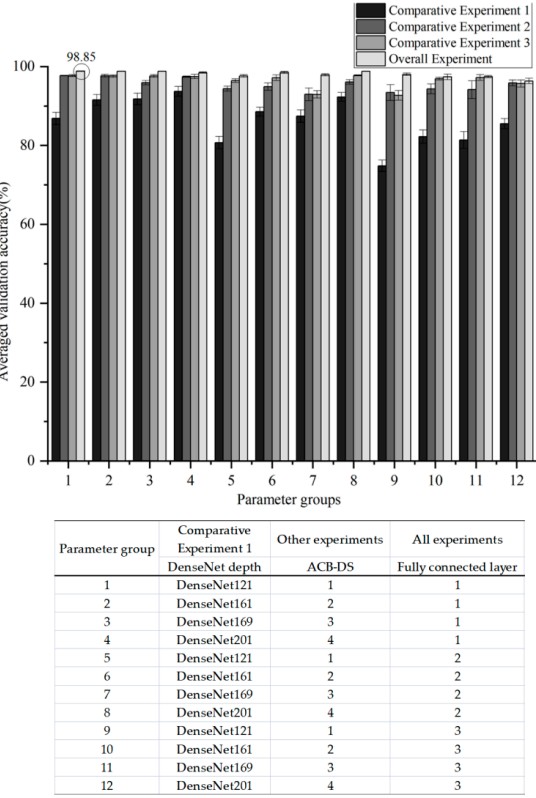

| Parameter group | Comparative Experiment 1 | Other experiments | All experiments |
|---|---|---|---|
| | DenseNet depth | ACB-DS | Fully connected layer |
| 1 | DenseNet121 | 1 | 1 |
| 2 | DenseNet161 | 2 | 1 |
| 3 | DenseNet169 | 3 | 1 |
| 4 | DenseNet201 | 4 | 1 |
| 5 | DenseNet121 | 1 | 2 |
| 6 | DenseNet161 | 2 | 2 |
| 7 | DenseNet169 | 3 | 2 |
| 8 | DenseNet201 | 4 | 2 |
| 9 | DenseNet121 | 1 | 3 |
| 10 | DenseNet161 | 2 | 3 |
| 11 | DenseNet169 | 3 | 3 |
| 12 | DenseNet201 | 4 | 3 |

**Figure 6.** Average and standard deviation values of the validation accuracy for the four groups of comparative experiments.

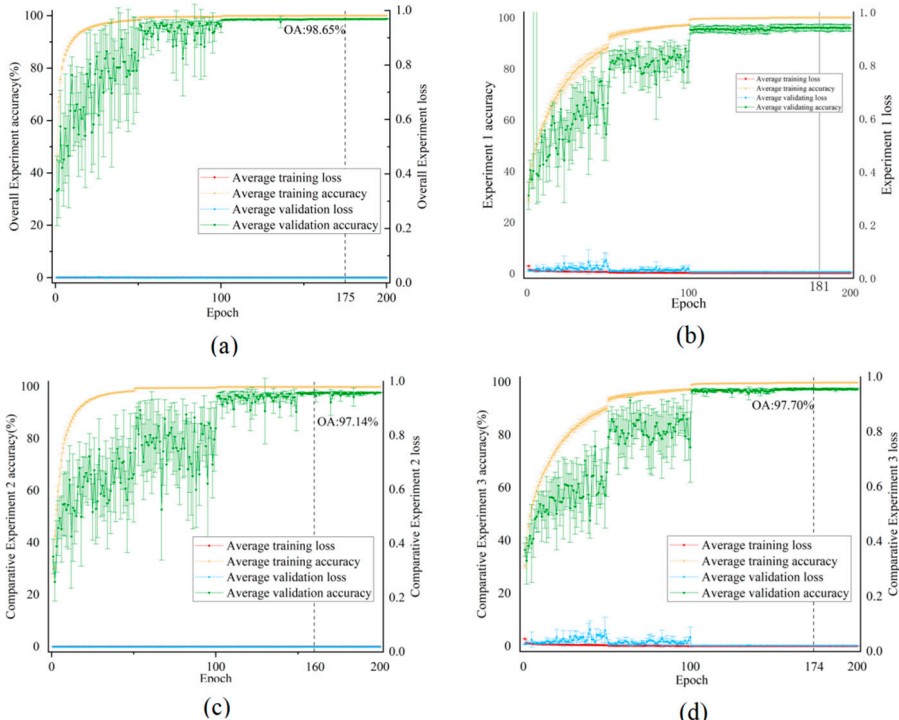

**Figure 7.** The average and standard deviation values of the training and validation accuracies and losses at each epoch for the Overall Experiment (**a**), Comparative Experiment 1 (**b**), Comparative Experiment 2 (**c**), and Comparative Experiment 3 (**d**).

For the Overall Experiment (Figure 7a), the model converged at an epoch of 175 and the OA reached 98.65%. Comparative Experiment 1 (Figure 7b) converged at an epoch of 178, and the OA was 89.10%. Comparative Experiment 2 (Figure 7c) converged at 160, and the OA was 94.83%. Comparative experiment 3 (Figure 7d) converged at 174, with an OA of 97.70%.

After the introduction of the ACB-DS block into DenseNet, the convergence speed of the network was markedly accelerated. Although the training speed of the network decreased after the addition of the BiLSTM module and low-level spectral–spatial features, it remained smaller than the convergence rounds presented in Figure 7b. The Overall Experiment conducted on the proposed network also converged at a relatively fast speed. Other comparative experiments have one or two step-by-step increases in both the training and validation accuracy curves, and the network may jump out of the local optima. The training and validation accuracy curve trends of the Overall Experiment were relatively flat, and local optimization was not easily achieved. The ACB-DS block significantly reduced the computational parameters. Furthermore, the proposed network does not introduce too many redundant modules. The addition of ACB-DS blocks and low-level spectral–spatial features can effectively reduce the starting point of the training and validation loss curves and result in a smoother curve trend.

Overall, although the 3S-A$^2$ DenseNet-BiLSTM model proposed in this study has a markedly more complex structure and a deeper network than DenseNet, it still rapidly converges. The training and validation losses decreased more smoothly than the remaining three sets of comparative experiments, and the training and validation accuracies of the remaining experiments displayed a very steep upward trend. Therefore, the 3S-A$^2$ DenseNet-BiLSTM model had a better performance and more stable properties.

*4.2. Results of Accuracy Assessment*

The different experiments had different performances in terms of the F1-score, kappa, and OA; the results are presented in Table 4. For the four groups of experiments, the OA values appeared in the order of 98.65% ± 0.05% (Overall Experiment), 97.70% ± 0.08% (Comparative Experiment 3), 97.14% ± 0.28% (Comparative Experiment 2), and 94.83% ± 0.04% (Comparative Experiment 1). The same trend was observed for the other metrics. The experimental results show that the 3S-A$^2$ DenseNet-BiLSTM method is effective for FLCCs in complex mining areas.

**Table 4.** The average and standard deviation values of the four groups of experiments on the metrics of F1-score, kappa, and OA (%).

| Experiment | F1-Score | Kappa | OA |
| --- | --- | --- | --- |
| Comparative Experiment 1 | 94.82 ± 0.04 | 94.56 ± 0.04 | 94.83 ± 0.04 |
| Comparative Experiment 2 | 97.13 ± 0.29 | 96.99 ± 0.30 | 97.14 ± 0.28 |
| Comparative Experiment 3 | 97.69 ± 0.08 | 97.57 ± 0.08 | 97.70 ± 0.08 |
| Overall Experiment | 98.65 ± 0.05 | 98.65 ± 0.05 | 98.65 ± 0.05 |

Compared with Comparative Experiment 1, after the introduction of the ACB-DS block and parallel attention mechanism in Comparative Experiment 2, the F1-score, kappa, and OA increased by 2.43%, 2.43%, and 2.31%. Compared with Comparative Experiment 2, after the introduction of the BiLSTM module in Comparative Experiment 3, the F1-score, kappa, and OA were increased by 0.56%, 0.58%, and 0.56%, respectively. Compared with Comparative Experiment 3, after introducing the low-level spectral–spatial feature and three-stream feature fusion strategy into the Overall Experiment, the F1-score, kappa, and OA increased by 1.08%, 1.08%, and 0.95%, respectively. The superior performance of the proposed model has been demonstrated. Our previous study [24] concluded that statistical testing depends on the size of the test set. Owing to the small sample size of the test set in this study, a statistical test was not performed. Test sets with different sizes and corresponding statistical tests will be investigated in future studies.

As shown in Table 5, the classification accuracies of each land cover type reached a high level. In fact, even the average F1-measures of woodland, white roof, fallow land, and shrubbery, with the worst classification effects, were above 87%.

**Table 5.** Average and standard deviation values of the F1-measures for the four groups of experiments (%).

| Class | Overall Experiment | Comparative Experiment 1 | Comparative Experiment 2 | Comparative Experiment 3 |
|---|---|---|---|---|
| Open pit | 99.75 ± 0.22 | 95.80 ± 0.20 | 97.90 ± 1.02 | 97.90 ± 0.50 |
| Ore processing site | 99.45 ± 0.09 | 92.92 ± 0.28 | 97.35 ± 1.17 | 98.20 ± 0.20 |
| Dumping ground | 99.70 ± 0.17 | 96.90 ± 0.10 | 99.20 ± 0.14 | 99.20 ± 0.00 |
| Paddy | 99.75 ± 0.09 | 96.20 ± 0.20 | 98.75 ± 0.70 | 99.00 ± 0.20 |
| Greenhouse | 100.00 ± 0.00 | 100.00 ± 0.00 | 99.55 ± 0.09 | 100.00 ± 0.00 |
| Green dry land | 99.35 ± 0.22 | 93.20 ± 2.00 | 96.85 ± 1.19 | 96.90 ± 0.10 |
| Fallow land | 96.90 ± 0.59 | 87.70 ± 0.10 | 91.80 ± 2.78 | 94.00 ± 0.60 |
| Woodland | 95.30 ± 0.30 | 90.80 ± 0.20 | 92.20 ± 1.07 | 93.00 ± 0.20 |
| Shrubbery | 92.35 ± 0.52 | 80.30 ± 0.70 | 86.90 ± 2.34 | 89.70 ± 0.10 |
| Coerced forest | 99.55 ± 0.09 | 96.20 ± 0.00 | 99.00 ± 0.45 | 99.00 ± 0.00 |
| Nursery | 98.20 ± 0.62 | 90.70 ± 2.10 | 96.85 ± 1.36 | 98.40 ± 0.20 |
| Pond and stream | 99.15 ± 0.41 | 95.40 ± 0.20 | 98.35 ± 0.57 | 99.10 ± 0.10 |
| Mine pit pond | 100.00 ± 0.00 | 99.90 ± 0.10 | 99.85 ± 0.09 | 99.80 ± 0.00 |
| Dark road | 100.00 ± 0.00 | 100.00 ± 0.00 | 99.70 ± 0.17 | 100.00 ± 0.00 |
| Light gray road | 99.70 ± 0.17 | 97.10 ± 0.50 | 98.50 ± 0.71 | 98.80 ± 0.20 |
| Bright road | 99.95 ± 0.09 | 98.70 ± 0.50 | 98.75 ± 0.80 | 99.60 ± 0.20 |
| Blue roof | 99.80 ± 0.00 | 99.10 ± 0.10 | 99.75 ± 0.09 | 99.70 ± 0.10 |
| White roof | 94.85 ± 0.43 | 91.80 ± 0.40 | 94.50 ± 1.68 | 93.40 ± 0.20 |
| Red roof | 99.75 ± 0.09 | 97.40 ± 0.40 | 99.55 ± 0.26 | 99.50 ± 0.10 |
| Bare surface land | 99.55 ± 0.17 | 96.50 ± 0.30 | 97.15 ± 2.17 | 98.70 ± 0.10 |

In Comparative Experiment 1, the F1-measures of most classes exceeded 94%, with seven exceptions: woodland, white roof, fallow land, green dry land, shrubbery, nursery, and ore processing sites. Among them, fallow land and shrubbery had the lowest F1-measures (lower than 90%). However, the accuracies for dark road and greenhouse in Comparative Experiment 1 reached 100%.

In Comparative Experiment 2, the F1-measures of most classes exceeded 97%, with only six exceptions: woodland, white roof, fallow land, green dry land, nursery, and shrubbery. Generally, the F1-measures of most classes in Comparative Experiment 2 were higher than those of Comparative Experiment 1. Only dark road, greenhouse, and mine pit pond lagged behind Comparative Experiment 1, with accuracy decreases of 0.30%, 0.45%, and 0.05%, respectively. Furthermore, the F1-measures of the other 17 classes were 0.05–6.60% higher than that of Comparative Experiment 1.

In Comparative Experiment 3, the F1-measures of most classes exceeded 97%, with only five exceptions. The accuracy of woodland, white roof, fallow land, and green dry land was still greater than 93%, and only shrubbery had an accuracy of 89.70%. Dark roads and greenhouses were correctly classified. Generally, most types of F1-measures in Comparative Experiment 3 were higher than those in Comparative Experiment 2. Blue roof, white roof, red roof, and mine pit pond lagged by accuracy decreases of only 0.05%, 1.10%, 0.05%, and 0.05%, respectively.

The 3S-A$^2$ DenseNet-BiLSTM method had the best classification effect for almost all 20 types, except for Comparative Experiment 3, which had the best classification effect for nurseries (with a narrow margin of 0.20%). In the Overall Experiment, the F1-measures of most classes exceeded 98%, and the classifier could correctly identify all dark roads, greenhouses, and mine pit ponds, leading to the same conclusion as in Section 4.1: the 3S-A$^2$ DenseNet-BiLSTM model has a significant effect on the FLCCs of complex mining areas.

### 4.3. Results of Visual Prediction

The four sub-images in Figure 8 clearly show the performances of the four experimental groups. These images comprise 20 different colors that represent 20 secondary land cover classes. Compared with the real image (Figure 1), the four prediction maps were found to be visually accurate. In particular, the visual prediction maps of Comparative Experiment 1 and the Overall Experiment show that the results of the visual prediction are quite different, ultimately confirming the results presented in Table 5. The best algorithm (Figure 8b) is not only better than the worst algorithm (Figure 8a), but is also better by a large gap. Ultimately, the 3S-A$^2$ DenseNet-BiLSTM model can be proven to have a superior performance.

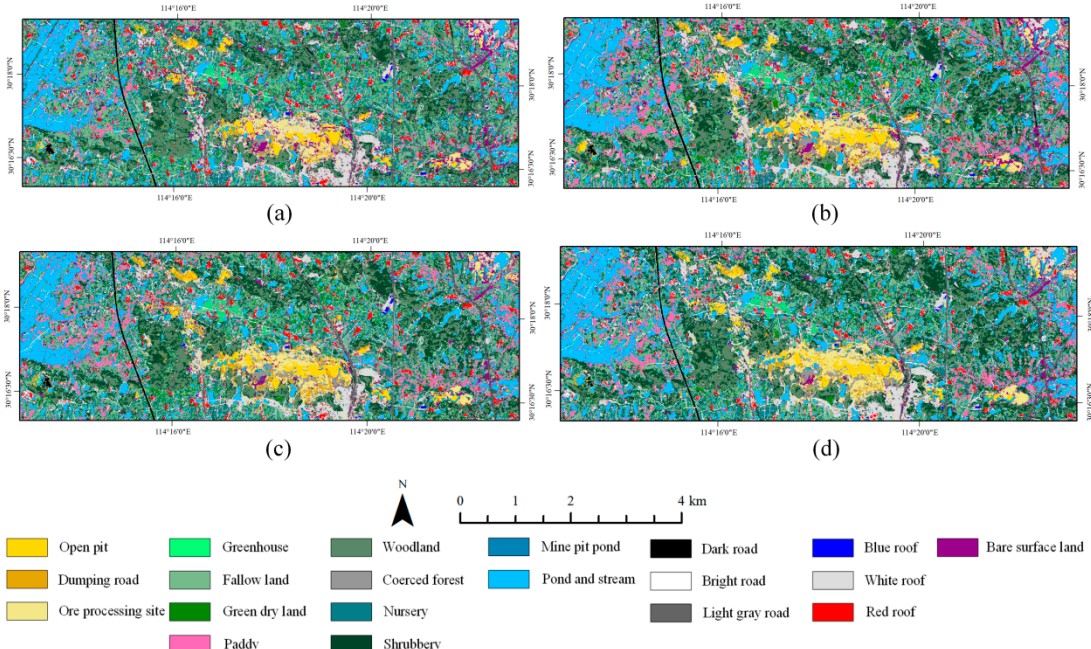

**Figure 8.** Visual prediction maps of the study area for Comparative Experiment 1 (**a**), Comparative Experiment 2 (**b**), Comparative Experiment 3 (**c**), and the Overall Experiment (**d**).

According to Figure 8d, the 3S-A$^2$ DenseNet-BiLSTM model obtained better classification results than the other three models. In particular, the open pit, light gray road, green dryland, ore processing site, all-dark road, greenhouse, and mine pit ponds features were correctly classified. The four groups of experiments had some classification errors for woodland, fallow land, nursery, and shrubbery. Notably, the features of these four types of ground objects might be similar; that is, the spectral–spatial and topographic differences are small. The reasons for this analysis are as follows.

(1) Compared with Comparative Experiment 1, Comparative Experiment 2 introduces a parallel channel, spatial attention mechanism, and ACB-DS block, which proves the accuracy of 20 classes of land cover on a large scale for the first iteration.
(2) Compared with Comparative Experiment 2, Comparative Experiment 3 adds BiLSTM based on A$^2$ DenseNet, allowing the model to extract correlation information among the spectral channels. Therefore, the classification accuracy of fallow land, woodland, and shrubbery with similar spectral features was markedly improved for the second iteration.
(3) Compared with Comparative Experiment 3, Comparative Experiment 4 introduces 106 dimensional low-level features and a multistream post-fusion strategy based on A$^2$ DenseNet-BiLSTM. The features that were manually extracted cover the shortage of depth features extracted by the depth model in detail and have a higher resolution. Therefore, the accuracy of the 20 land cover classes was improved to a certain extent.

## 5. Discussion

*5.1. Effectiveness of the Proposed 3S-A$^2$ DenseNet-BiLSTM Model*

According to the results presented in Section 4, the 3S-A$^2$ DenseNet-BiLSTM model has the best FLCC performance for complex mining areas relative to the remaining three classification models proposed in this study.

Many studies have confirmed the effectiveness of the various network modules and fusion strategies introduced in this study. For the ACB-DS module, Lo et al. [53] proposed an efficient dense module with an ACB and dilated convolution structure, which was 2.7 times faster than the comparison network. According to Wang et al. [54], the spatial symmetry of existing convolutional blocks limits the ability of the network to learn the spatial location of features. Thus, a multiscale, spatially asymmetric recalibration network was designed to extract feature maps with spatial asymmetry. Zhu et al. [55] used ACB to reduce the number of floating-point operations and parameters in the hyperspectral image classification process, and simultaneously solved the problem where traditional convolution blocks could not capture features with arbitrary sizes and shapes. Wang et al. [56] designed a spectral and spatial SENet for channel and spatial attention mechanisms, which was used to model the interdependencies between channels and spaces to recalibrate feature responses, with good performance in HSI classification. Roy et al. [57] and Roy et al. [58] used parallel spatial and channel SENet modules to extract richer spatial and channel features. Zhang et al. [59] introduced SENet after each convolutional layer to construct a channel-feature reweighted DenseNet. According to Zhang et al. [60], the convolution operation would ignore the influence of feature points far away from the current region and thus used SENet for feature point channel recalibration. For the multistream feature fusion strategy, Liu et al. [61] used a two-stream CNN to extract the spectral–spatial information of HSI and a cascade network to extract the spatial information of multispectral imagery, forming an effective three-stream network. Ge et al. [62] utilized a multistream 2D CNN to extract features from multi-source datasets. Based on the above studies, the modules and strategies used in the 3S-A$^2$ DenseNet-BiLSTM method are important for improving the accuracy of the FLCCs. The detailed analyses were as follows:

(1) 3S-A$^2$ DenseNet-BiLSTM can learn the joint representation of low-level spectral–spatial, deep spectral–spatial, and deep topographic features. It can also capture the complete information of different emphases of the same object landscape under different imaging methods that cannot be perceived by a single form of data.

(2) Compared with the classic DenseNet method, the A$^2$ DenseNet module used the ACB-DS block and a double attention mechanism to extract more discriminative features. Simultaneously, the number of parameter calculations was reduced, and the speed of network convergence was accelerated. As the spatial regions of various ground objects are not always symmetrical, traditional symmetric convolution was not suitable for extracting the features of irregularly shaped land covers. However, the ACB-DS method solved the problem well, and was able to extract richer and more detailed spatial information.

(3) BiLSTM was used to model the contextual features of correlations across different channels. It supplemented the global features extracted by A$^2$ DenseNet and outputted more abundant spectral–spatial and topographic features, respectively.

(4) A multistream post-fusion strategy was used to further fuse the low-level spectral–spatial features and multimodal deep features extracted by the A$^2$ DenseNet-BiLSTM model. This strategy took full advantage of the spectral, spatial, and topographic information to obtain joint representations.

This study had some limitations in the broadest context:

(1) The model training time was too long. During the performance of the FLCC experiments, each model training process took approximately 30 h, which is unfavorable for the real-time monitoring of mining activities. A large amount of time hinders the process of industrializing the research. To assess the complexity of the model,

both the performance of the model and the difficulty of training must be considered. On the one hand, the model must be as accurate as possible, which requires it to have a higher expression ability; therefore, it is easier to achieve this goal using a model with a higher complexity. On the other hand, if the complexity of the deep model is too high, it will increase the difficulty of training, thus wasting computing resources. Therefore, reducing the training time as much as possible without reducing the expression ability of the deep model requires further research.

(2) An insufficient sample size was employed. For the data used for model training, only 2000 sample points were used for each class of land cover. For more complex deep models, the nonglobal features of the training data can be learned by the model. However, in the field of remote sensing, it is usually difficult and costly to obtain labeled data; thus, reducing the complexity of the model without reducing its classification accuracy requires further research.

(3) The generalization performance is unknown. Normally, many illegal mining activities may be carried out at night; however, the current study only monitored mining activities during the day. The performance of the model on remote sensing satellite images at night still requires further transfer learning research.

### 5.2. Effects of Different Models

To validate the performance of the proposed model, comparisons were made based on the results of previous studies [7,23,24] (Table 6), including traditional MLAs (i.e., RF, SVM, and feature subset-based SVM (FS-SVM)) and some DL methods, such as multilevel output-based DBN (DBN-ML), DBN-RF, and CNN.

**Table 6.** Accuracy evaluation results of the 13 classification models on the ZY-3 dataset (%).

| Model | F1-Score | Kappa | OA | Description |
|---|---|---|---|---|
| 3S-A2 DenseNet-BiLSTM | 98.65 ± 0.05 | 98.65 ± 0.05 | 98.65 ± 0.05 | The proposed network |
| Single-scale CNN [7] | | | 93.76 ± 0.76 | Multimodal and single-scale kernel-based multistream CNN |
| 3M-CNN [7] | | | 95.11 ± 0.48 | Multimodal and multiscale kernel-based multistream CNN |
| 3M-CNN-Magnify [7] | | | 96.60 ± 0.22 | Multimodal and multiscale kernel-based multistream CNN with the selected parameter value |
| DBN-ML [24] | 95.07 | 94.84 | 95.10 | Multi-level output-based deep belief network |
| RF [23] | 88.85 ± 0.22 | 88.31 ± 0.22 | 88.90 ± 0.20 | Random forest |
| SVM [23] | 77.79 ± 0.54 | 76.72 ± 0.55 | 77.88 ± 0.53 | Support vector machine |
| FS-SVM [23] | 91.75 ± 0.57 | 91.34 ± 0.60 | 91.77 ± 0.57 | SVM with feature fusion method |
| DBN-S [23] | 94.22 ± 0.67 | 93.93 ± 0.70 | 94.23 ± 0.67 | DBN with Softmax classifier |
| DBN-RF [23] | 94.05 ± 0.34 | 93.76 ± 0.36 | 94.07 ± 0.34 | DBN with feature fusion method |
| DBN-SVM [23] | 94.72 ± 0.35 | 94.46 ± 0.37 | 94.74 ± 0.35 | DBN with SVM classifier |
| CNN [24] | 90.15 ± 1.66 | 89.68 ± 1.75 | 90.20 ± 1.64 | VGG network |
| DCNN [24] | 95.00 | 94.76 | 95.02 | VGG with deformable convolutions |

For the models based on MLAs, the OA values decreased for RF (88.90% ± 0.20%) [23], SVM (77.88% ± 0.53%) [23], and FS-SVM (91.77% ± 0.57%) [23]. Kwan et al. [63] demonstrated that SVM can achieve better LCC results, and Goldblatt et al. [64] found that the RF model using Landsat 8 satellite data performed better than when using Landsat 7 data. However, when the normalized vegetation and building indices were added to the Landsat 7 data, the performance improvement of the SVM was the most significant.

The DL-based models have been demonstrated to be significantly better than traditional MLAs. The order of OA values observed was: multimodal remote sensing data and multiscale kernel-based multistream CNN with the selected value of parameter k (3M-CNN-Magnify) (96.60%) [7]; multimodal remote sensing data and multiscale kernel-based multistream CNN (3M-CNN) (95.11%) [7]; DBN-ML (95.10%) [24]; DCNN (95.02%) [24]; DBN-SVM (94.74% ± 0.35%) [24]; DBN-S (94.23% ± 0.67%) [23]; DBN-RF (94.07% ± 0.34%) [23]; single-scale CNN(93.76% ± 0.76%) [7]; and CNN (90.20% ± 1.64%) [24]. Compared with the best MLA of FS-SVM, all deep models had better classification performances. Among them, the best model, 3M-CNN-Magnify, had an improved OA of 4.83%. Other studies involving DL-based models have reached similar conclusions. For example, Zhao et al. [65] proved that CNN can effectively handle rice mapping in complex landscape areas. Several researchers have also proposed DBN-based models. For example, Li et al. [66] proposed a multilabel electrical signal classification method based on DBN-RF, which had better classification accuracies and recognition efficiencies than the naive Bayes, k-nearest neighbor (KNN), SVM, and RF models. By comparing DBN-S with the classical KNN and SVM algorithms, Jiang et al. [67] found that the proposed hybrid algorithm can achieve more satisfactory results. Wang et al. [68] demonstrated that the SVM algorithm outperformed single classification methods that used DBN, SVM, and maximum likelihood estimation. Meanwhile, Li et al. [24] revealed that DBN-ML can produce better results in multiscale feature extraction for complex surface conditions while effectively solving the overfitting problem of the DBN models.

The proposed 3S-A$^2$ DenseNet-BiLSTM not only achieved the highest classification accuracy among the comparison models as demonstrated in Figure 9, but also achieved the visual effect closest to the real satellite image (Figure 1), especially for the open pit, bare surface land, and other classes. As shown in Figure 9a–c, they have been misclassified to varying degrees. DBN-S and DBN-SVM also misclassified dark roads as bare surface land and open pits. Notably, none of the models were suitable for the classification of fallow land, woodland, and shrubbery. The reasons for this are summarized as follows:

(1) Open pit, bare surface land, fallow land, woodland, and shrubbery have relatively similar spectral features, but neither the DBN-based model nor the 3M-CNN-Magnify considers the contextual correlation among spectral bands. Accordingly, it is impossible to distinguish between land covers with similar spectral features and visually similar colors.

(2) The DBN-based model classifies dark roads into land covers that are similar to this class in terms of geographical space. Although they do not have similar spectral features, the dark road passes through bare surface land and open pits. 3M-CNN-Magnify uses a multiscale kernel-based convolution block, which can select different convolution kernel sizes according to the multi-size data input, making full use of spatial neighborhood information. The proposed 3S-A$^2$ DenseNet-BiLSTM model introduces an ACB-DS block, which can extract the features of irregularly shaped land covers, making the extracted spatial features more abundant. However, DBN-S and DBN-SVM do not have a structure that is more conducive to the extraction of spatial features; thus, they cannot distinguish land covers with strong spatial correlation.

(3) All models confuse the three classes of land cover: fallow land, woodland, and shrubbery; this may be due to the small spectral differences between them. Moreover, it is difficult to perform visual distinction; however, their height difference is large. Therefore, the model can only extract the discriminative features according to the topographic height difference of the DEM data. However, the 3S-A$^2$ DenseNet-BiLSTM model only extracts the neighborhood size of topographic data instead of a complete image, which may weaken the advantage of topographic information.

In summary, the proposed 3S-A$^2$ DenseNet-BiLSTM model yielded better performance for the FLCCs of complex mining areas.

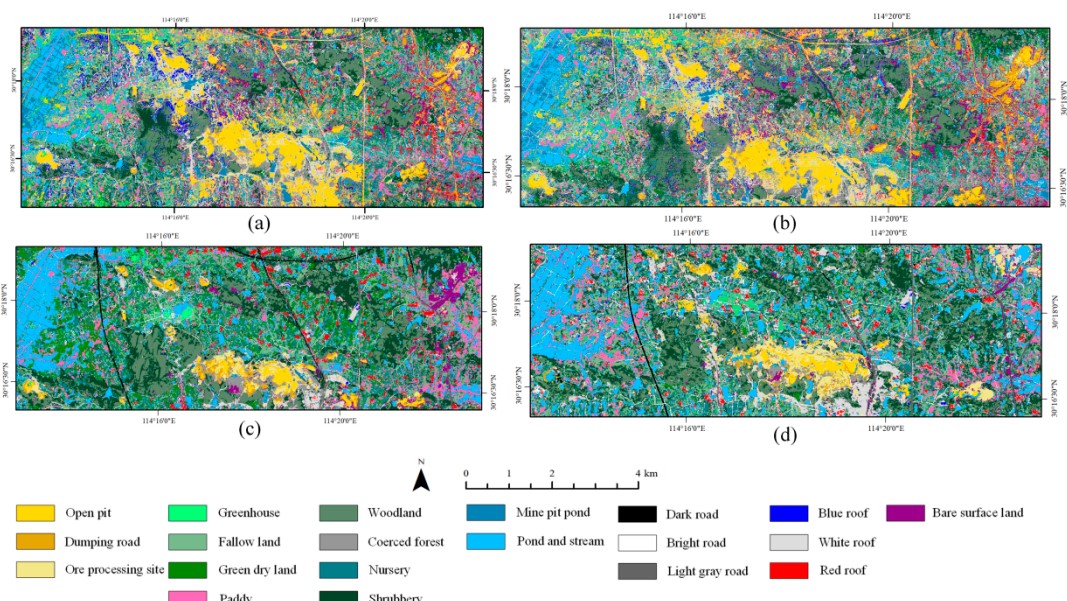

**Figure 9.** Visual prediction maps of the study area for the DBN-S (**a**), DBN-SVM (**b**), 3M-CNN-Magnify (**c**), and 3S-A$^2$ DenseNet-BiLSTM (**d**) models.

## 6. Conclusions

In this study, a three-stream double attention network, 3S-A$^2$ DenseNet-BiLSTM, was proposed for the FLCC of complex mining areas.

Four sets of experiments were performed, and the proposed model was proven to have the best classification performance. Compared with the basic model, the proposed model had an overall accuracy of 98.65% $\pm$ 0.05% and obtained better results on the visual prediction map, with an increase in accuracy of 4.03%. Accordingly, the proposed model facilitates the study of complex landscapes. The following conclusions were drawn: (1) the three-stream multimodal feature learning and post-fusion method was effective for learning and fusing low-level spectral–spatial, deep spectral–spatial, and topographic features; (2) integrating ACB-DS blocks into DenseNet can extract richer spatial information; (3) the double attention mechanism can adaptively select more important features; and (4) BiLSTM can extract cross-channel context features.

In conclusion, the proposed 3S-A$^2$ DenseNet-BiLSTM model was determined to be effective for the FLCC of complex mining areas. In the future, we will apply the main study area algorithm to other auxiliary study areas and carry out temporal and spatial transfer learning of the proposed model with the newest remote sensing data and different remote sensing satellite imagery such as Sentinel-3, Landsat, Sentinel-2, HJ-1A, and Gaofen-7 to test the generalization capability of the model. Meanwhile, we will attempt to use the proposed model for the fine classification of other complex geographic environments, as well as build related large remote sensing datasets.

**Author Contributions:** Conceptualization, D.Z., X.L. and W.H.; Data curation, J.L. and W.C.; Investigation, D.Z., J.L., X.L., W.H. and W.C.; Methodology, D.Z., J.L., X.L., W.H. and W.C.; Software, D.Z. and J.L.; Validation, D.Z. and J.L.; Visualization, D.Z., J.L., X.L., W.H. and W.C.; Writing—original draft, D.Z. and J.L.; Writing—review & editing, D.Z., J.L., X.L., W.H. and W.C. All authors have read and agreed to the published version of the manuscript.

**Funding:** This research was funded by National Natural Science Foundation of China: 42071430; Department of Natural Resources of Hubei Province: ZRZY2021KJ04; National Natural Science Foundation of China: U21A2013; College Students' Innovation and Entrepreneurship Training Program: S202110491130. And The APC was funded by National Natural Science Foundation of China: 42071430.

**Conflicts of Interest:** The authors declare no conflict of interest.

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
