# Peer review of "Three-Stream and Double Attention-Based DenseNet-BiLSTM for Fine Land Cover Classification of Complex Mining Landscapes"

_sustainability, doi:10.3390/su141912465_

Round 1

Reviewer 1 Report

For the task of fine land cover classification, this manuscript proposed a three-stream double attention network by combining the DenseNet and bidirectional long short-term memory network. The idea is interesting and the result is basically satisfactory. However, some other problems in the manuscript are still concerned in the following:
1. The organization of this manuscript should be added to the end of the
introduction.
2. In Figure 2, please show and include the result of image classification.
3. In the experiments, could the authors also show the visual results of different
models?
4. In order to improve the readability, some unnecessary abbreviations are suggested to be deleted.
5. The conclusions could be simplified.
6. The experiment setting should be introduced.
7. For deep learning based works, more related models on GAN could be included in the introduction, such as “DOI: 10.1109/LGRS.2020.3009017”, “DOI: 10.1016/j.isprsjprs.2021.07.007”…

Reviewer 2 Report

Dear Author(s);

please kindly check the attached file.

Reviewer 3 Report

General comments:

   This study conducted a three-stream double attention network, 3S-A2 DenseNet-BiLSTM and tried to mine the spectral-spatial and topographic information of multimodal ZY-3 data and their contextual interdependence. It can be seen that the author has a deep foundation in related fields. Thus, the structure of the methods, especially the architecture of machine learning, were clearly showed in this manuscript. Also, the author showed an effective performance of 3S-A2 DenseNet-BiLSTM. 
However, before it was published, I think the author should consider and address the following two questions.

Detailed comments: 

1.       Introduction, the key question I worry about. The author seems accustomed to give amounts of published papers examples, as “WHO found…, other found …, the rest found…”, but the internal logic is not very prominent. For examples, Lines 52-59; Lines 99-104; Lines 117-123.
Besides, pay attention to the continuity between paragraphs. For example, Alternatively, I suggest, take the third paragraph as the general logic (Lines 64-91), revise the rest paragraphs in the introduction section, improve the logic expression of these literatures.
2. Results of accuracy assessment: the results from different methods are significant? In other words, did the DenseNet-BiLSTM in this study significantly improve the model performance? This may need further supplementary analysis.

Reviewer 4 Report

The manuscript is good, the authors evaluate the ‘’3S-A2 DenseNet-BiLSTM: A Three-stream Double Attention Network for Fine Land Cover Classification of Complex Min-3 ing Landcapes’’. It is an interesting and great contribution to the scientific community; however, the results, discussion and references of the paper should be improved. Still there are many issues present in the manuscript which should be explained properly. The manuscript needs some minor revisions as given below:

·         The text of this paper in general needs a thorough review, as there are multiple spelling and grammatical errors. Many sentences do not mean any sense. Moreover, there are several sloppy errors that should be fixed.

·         Abstract is too short, write some sentence about your results in abstract.

·         Please use similar font size and font style throughout.

·         More research background and motivation should be added to the Introduction section. Although, I propose some new papers can be added in the reference list and text which will also help you to make it more intriguing such

https://doi.org/10.1007/s11356-022-21650-8,

https://doi.org/10.1038/s41598-022-17454-y

·         Crosscheck all the references you cited in the text and construct it according to journal guidelines.

·         Line 38, Give suitable reference of this sentence.

·         Line 160, in this study, you used satellite image for the year 2012. I think this data set id very old, please use latest data for analysis.

·         Discussion: As per the instruction given by the journal “The findings and their implications should be discussed in the broadest context possible and the limitations of the work highlighted”.

·         The conclusion is too general. What are the key findings of this study?

·         Write the future recommendation of your research in conclusion.

Overall, the study conducted is interesting but a minor revision of the entire manuscript is essentially required for publication in this journal. Hence, I recommend reconsideration after a minor revision of the manuscript.  

Round 2

Reviewer 2 Report

Dear author(s);

The main queries are addressed in the text.

Author Response

The revised manuscript has been checked by a native English speaker. Please see if the revised version met the English presentation standard.